# Linking Surface Facts to Large-Scale Knowledge Graphs

**Gorjan Radevski**[1,2], **Kiril Gashteovski**[1,3], **Chia-Chien Hung**[1],
**Carolin Lawrence**[1], **Goran Glavaš**[4]

[1]NEC Laboratories Europe, Heidelberg, Germany; [2]KU Leuven, Leuven, Belgium;
[3]CAIR, Ss. Cyril and Methodius University, Skopje, North Macedonia;
[4]CAIDAS, University of Würzburg, Würzburg, Germany
[1]firstname.lastname@neclab.eu, [2]gorjan.radevski@kuleuven.be,
[4]goran.glavas@uni-wuerzburg.de

## Abstract

Open Information Extraction (OIE) methods extract facts from natural language text in the form of (*"subject"*; *"relation"*; *"object"*) triples. These facts are, however, merely surface forms, the ambiguity of which impedes their downstream usage; e.g., the surface phrase *"Michael Jordan"* may refer to either the former basketball player or the university professor. Knowledge Graphs (KGs), on the other hand, contain facts in a canonical (i.e., unambiguous) form, but their coverage is limited by a static schema (i.e., a fixed set of entities and predicates). To bridge this gap, we need the best of both worlds: (i) high coverage of free-text OIEs, and (ii) semantic precision (i.e., monosemy) of KGs. In order to achieve this goal, we propose a new benchmark with novel evaluation protocols that can, for example, measure fact linking performance on a granular triple slot level, while also measuring if a system has the ability to recognize that a surface form has no match in the existing KG. Our extensive evaluation of several baselines shows that detection of out-of-KG entities and predicates is more difficult than accurate linking to existing ones, thus calling for more research efforts on this difficult task. We publicly release all resources (data, benchmark and code)[1].

## 1 Introduction

Open Information Extraction (OIE) methods extract surface *("subject"; "relation"; "object")-* triples from natural language text in a schema-free manner (Banko et al., 2007). For example, given the sentence *"Michael Jordan, who grew up in Wilmington, played for Chicago Bulls"*, an OIE system should extract the triples (i.e., *surface facts*): $t_1 =$(*"Michael Jordan"; "played for"; "Chicago Bulls"*) and $t_2 =$(*"Michael Jordan"; "grew up in"; "Wilmington"*). The output of such systems is used in a diverse range of downstream tasks, including

summarization (Ribeiro et al., 2022), question answering (QA) (Wu et al., 2022), event extraction (Dukić et al., 2023), text clustering (Viswanathan et al., 2023) and video grounding (Nan et al., 2021). OIE triples, however, consist of surface-form entities and relations, which are frequently ambiguous (e.g., in $t_1$ and $t_2$, the entity mention *"Michael Jordan"* may refer to several entities, e.g. the basketball player or the computer scientist). Resolving such ambiguities, by linking the OIE triple slots to inventories of unambiguous concepts, improves their downstream utility, e.g., in QA (Saxena et al., 2020), automatic medical diagnosing (Li et al., 2022) or dialogue (Joko et al., 2021).

Knowledge Graphs (KGs), on the other hand, are inventories of *canonical* facts in the form of (subject; predicate; object)-triples, where each slot is a unique (i.e., unambiguous) concept (Vrandečić, 2012). KGs are, however, limited by their own static and often hand-crafted schema (i.e., fixed set of entities and predicates). As a consequence, methods that directly extract canonical KG triples from text (Trisedya et al., 2019; Josifoski et al., 2022), preemptively discard any information outside of the reference KG schema. Acknowledging this, Ye et al. (2023) recently proposed *schema-adaptable KG construction*, with the goal of extracting information for a KG with an evolving schema. Ye et al. (2023) conclude that OIE methods indeed extract meaningful new knowledge for such KGs, however, they point precisely to the ambiguity of the surface forms as the major obstacle.

To combine the best of both worlds, we need to bridge the gap between the schema-free (but ambiguous) surface facts extracted from text and the schema-fixed (but unambiguous) KG knowledge. However, existing benchmarks and models only partially address the problem. One line of work (Zhang et al., 2019; Jiang et al., 2021; Wood et al., 2021) assumes a setting, which is arguably unrealistic, because the OIE entity slots are *a priori*

---

[1]https://github.com/nec-research/fact-linking

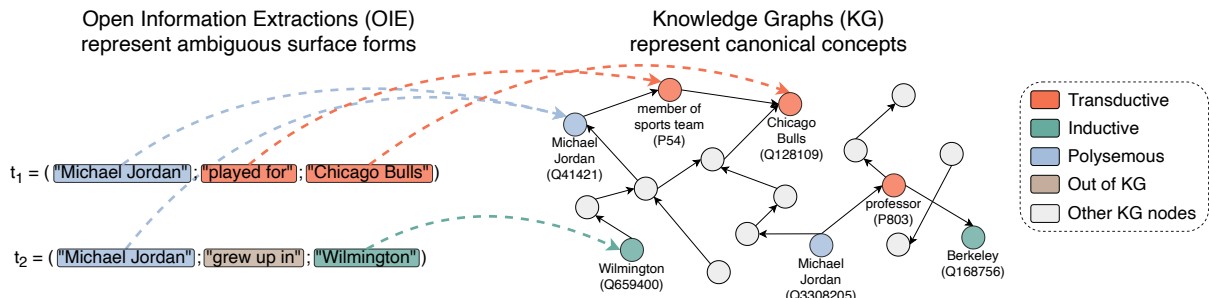

Figure 1: Open Information Extractions (OIEs) represent free-text surface-form triples, which may be ambiguous. Knowledge Graphs (KGs) represent canonical, unambiguous concepts, yet are limited by a hand-crafted schema. By linking OIEs to KG facts we bridge the gap between the schema-free (but ambiguous) surface facts extracted from text and the schema-fixed (but unambiguous) KG knowledge. Our benchmark addresses *all* aspects of the OIE-to-KG linking. In **transductive** evaluation, the KG facts consist of entities *seen* during training, while in **inductive** evaluation the KG facts contain entities *strictly outside* of the training data. **Polysemous** OIEs match to *multiple* KG concepts (e.g., "*Michael Jordan*"). Finally, some OIEs might refer to **Out-of-KG** entities or predicates.

linked to KG entities and thus only addresses relation linking. A second line of research (Trisedya et al., 2019; Cabot and Navigli, 2021; Josifoski et al., 2022; Sakor et al., 2020; Elsahar et al., 2018) entails canonical KG triples directly from text paragraphs, bypassing OIE, which makes such models tied to the fixed KG schema. Critically, *all* these works conjecture that *each* OIE slot *has* a corresponding KG entry. Gashteovski et al. (2019) showed that this is hardly ever the case in practice.

**Contributions and Findings.** We move away from the unrealistic and incomplete assumptions of prior work and propose ① a novel large-scale benchmark for OIE-to-KG linking; ② multifaceted evaluation protocols that cover *all* aspects of linking OIE facts to KGs (for an overview, see Fig. 1); ③ several strong baselines, inspired by state-of-the-art entity linking (Wu et al., 2019) and cross-modal retrieval (Miech et al., 2021; Geigle et al., 2022) approaches.

Through our experimental study, we found that the methods (i) perform well *transductively* but (ii) their performance deteriorates in an *inductive* evaluation. Further, we find that (iii) a dedicated OIE-to-KG fact re-ranking model improves the linking performance of both inductive and polysemous OIEs, and that (iv) we obtain high performance by training models solely on a synthetic variant of our dataset (i.e., with the KG as the only human-annotated data). Lastly, we investigate the largely underexplored issue of detecting *Out-of-Knowledge-Graph* extractions. We show that (v) it is possible to detect Out-of-KG entities to an extent, however, the same does not hold for predicates: a task that our experiments identify as a difficult open problem, which requires more research attention.

## 2 Fact Linking: Problem Statement

For a given surface-form OIE triple $t_1 = ($"$s$"; "$r$"; "$o$"$)$, the goal is to link each slot to a canonical concept in a KG (*if* the corresponding concept exists in the KG): "$s$" $\rightarrow e_1 \in \mathcal{E}$; "$r$" $\rightarrow p \in \mathcal{P}$; "$o$" $\rightarrow e_2 \in \mathcal{E}$, with $\mathcal{E}$ and $\mathcal{P}$ as the (fixed) sets of KG entities and predicates. Importantly, our problem definition (and consequently evaluation) focuses on linking at the *fact level*, where each OIE slot is contextualized with the other two OIE slots.[2]

Additionally and crucially, we want linking models that can assign an empty set to OIE surface forms (e.g., "$s$" $\rightarrow \emptyset$) when they refer to concepts not present in the KG (i.e., *out-of-KG* concepts). To enable a realistic setup for linking free-text OIE triples to a KG, we build a benchmark which considers four different facets (see Fig. 1).

**Transductive Fact Linking.** In transductive linking, we measure how well the models link OIEs to KG facts consisting of entities and predicates seen during training (as components of training KG facts). Note that the testing KG facts (as whole triples) are not in the training data. Consider, for example, the extraction $t_1$ in Fig. 1. In the transductive linking task, the mentions *"played for"* and *"Chicago Bulls"* refer to the KG predicate (P54) and entity (Q128109) respectively; both seen by the model as part of other training KG facts. However, the whole triple (Q41421; P54; Q128109) to which

---

[2]Alternatively, additional context can be included, e.g., the provenance from which the OIE surface fact is obtained.

$t_1$ is linked was not part of the training data.

**Inductive Fact Linking.** The inductive setup evaluates the linking to KG facts that consist of entities that are not seen during the training of the models. In other words, this setup tests the generalization of fact linking models over entities. In Fig. 1, given $t_2$ as a test instance, the OIE entity *"Wilmington"* is inductive as it refers to a KG entity (Q659400) that is not part of any training KG fact.

**Polysemous Fact Linking.** We focus on OIEs for which the *"s"* and *"o"* slots are ambiguous w.r.t. the KG, i.e., in isolation they refer to a set of KG entities rather than a single entity. The mention *"Michael Jordan"* from either $t_1$ and $t_2$ (Fig. 1), in isolation, refers to both the basketball player (Q41421) and the computer scientist (Q3308205). Here, the other two OIE slots offer the disambiguation signal that is necessary for successful linking.

**Out-of-KG Detection.** We introduce a novel *out-of-KG detection* task, in which the models are to recognize that an OIE triple component (i.e., *"s"*, *"r"* or *"o"*) cannot be linked because they do not have a corresponding KG concept (e.g., the relation *"grew up in"* from the triple $t_2$ in Fig. 1).

## 3 FaLB: Fact Linking Benchmark

We set up an automatic data processing pipeline to derive an OIE-to-KG fact linking benchmark, which supports all four evaluation facets from §2. We refer to both the process (i.e., pipeline) and the resulting benchmark as **FaLB**. FaLB's input is a dataset with (gold) alignments between natural language sentences and KG facts entailed by the sentence; i.e., each data instance is *(sentence, KG fact)* pair. Consequently, the creation of FaLB entails five design decisions: selection of (i) sentence-to-KG fact dataset(s), and (ii) a reference KG; (iii) generating OIE triples, (iv) high-precision OIE-KG fact alignments, and (v) a data augmentation strategy to increase the diversity of the data. Below is an example instance of the FaLB dataset:

---

**Example Data Instance from FaLB**

**Sentence:** *"M. J., who was born in Brooklyn, played for the Bulls."*

**OIE surface facts:** $t_1$ =*("M. J."; "played for"; "the Bulls")* and $t_2$ =*("M. J."; "was born in"; "Brooklyn")*

**KG canonical fact identifiers:** (Q41421; P54; Q128109); (Q41421; P19; Q18419)

**KG text facts:** (Michael Jordan; member of sports team; Chicago Bulls); (Michael Jordan; place of birth; Brooklyn)

**KG entity aliases:** (Q41421: Air Jordan, Michael Jeffrey Jordan, His Airness); (Q128109: Bulls, The Bulls), ...

---

**Sentence-to-KG Fact Datasets.** We build FaLB benchmarks for two such existing datasets: REBEL (Cabot and Navigli, 2021) and SynthIE (Josifoski et al., 2023). *REBEL* is built from Wikipedia abstracts, where the manually hyperlinked entities in each sentence are linked to Wikidata entities, thus making them golden. To match the sentence with a KG fact, for each pair of KG entities $e_i$ and $e_j$ within the sentence, REBEL obtains all KG predicates $p_k$ such that $(e_i, p_k, e_j)$ or $(e_j, p_k, e_j)$ exist. However, two Wikidata entity nodes connected with a predicate (thus constituting a KG fact) do not guarantee the predicate validity in the sentence. For that reason, Cabot and Navigli (2021) use a Natural Language Inference pre-trained RoBERTa (Liu et al., 2019) to filter the predicates *not entailed* by the Wikipedia sentence; see (Cabot and Navigli, 2021) for details. We also use *SynthIE*, which is synthetically generated using a Large Language Model (LLM) (Brown et al., 2020). Given a set of KG facts, the LLM is prompted to generate a sentence that covers the KG facts.[3]

**Reference KG.** We use Wikidata (Vrandečić, 2012) as our reference KG, because REBEL and SynthIE align sentences to Wikidata facts. We select the subgraph of Wikidata that contains all entities which have a corresponding Wikipedia page, as per Wu et al. (2019). Following Lerer et al. (2019), we additionally filter out the most infrequent entities and predicates, appearing less than 5 times in the whole Wikidata dump. This results in a large reference KG with $5,794,782$ unique entities and $4,153$ unique predicates. We further create two smaller, dataset-specific reference KGs; one for each of the two datasets for which we apply FaLB: REBEL and SynthIE. These Benchmark-Restricted KGs (BRKGs) contain only the Wikidata entities and predicates referenced by at least one OIE triple extracted from their respective datasets. The BRKG for REBEL contains $625,125$ entities and $565$ predicates, while the one for SynthIE contains $702,334$ entities and $758$ predicates.

**Generating OIE Triples.** We use four state-of-the-art OIE methods to obtain a set of OIE triples for each of the sentences in the dataset. To increase diversity, we use two state-of-the-art rule-based OIE methods: MinIE (Gashteovski et al., 2017) and StanfordOIE (Angeli et al., 2015); and two

---

[3]Josifoski et al. (2023) manually evaluated that SynthIE is of higher quality than REBEL (see Appendix D for details).

state-of-the-art neural OIE models: MilIE (Kotnis et al., 2022c) and Multi²OIE (Ro et al., 2020).

**High-precision OIE-KG Fact Alignments.** Next, we need to match the extracted OIE triples against the set of KG facts associated with the sentences. Crucially, this automatic step needs to have a high-precision in order to create a high quality benchmark. As per the distant supervision assumption of Mintz et al. (2009), we create an alignment $t \leftrightarrow f$ between any OIE triple $t$ and any KG fact $f$ that exactly match (i.e., have identical text form) in subject and object, assuming that the relation of $t$ is aligned with the predicate of $f$. When multiple OIE triples (excluding exact duplicates) $t_1$, $t_2$, ..., $t_k$ (e.g., extracted with different OIE systems) match with the same KG fact $f$, we obtain all $k$ alignments: $t_1 \leftrightarrow f$, ..., $t_k \leftrightarrow f$. Finally, we remove all training alignments $t \leftrightarrow f$ that exist in the test portion. To verify that this strategy indeed has high precision, we randomly sample 100 test instances, and conduct a human-study with two expert annotators. The annotators found 97% correct pairings (inter-annotator agreement of 99%; Kohen's kappa of 0.80); see Appendix F for details. We therefore confirm the reliability of this design choice to produce high quality data.

**Data Augmentation to Increase Diversity.** Lack of example diversity—most linked OIE entity mentions exactly match the text of the corresponding KG entities—is a common problem in existing linking benchmarks (Cabot and Navigli, 2021). This stems from strict data curation procedures that aim for high alignment precision, representing a mismatch with practice, where often the entity mentions do not exactly match the canonical KG entity denotation. To train and evaluate fact linking methods on more complex linking examples (e.g., linking from initials and abbreviations), we augment the data using the additional information available in the reference KG. For each alignment $t \leftrightarrow f$, we fetch the Wikidata aliases (i.e., non-canonical denotations) for the subject and object entities of $f$. We then create additional pairs $t' \leftrightarrow f$ with augmented OIE triples $t'$ for all possible alias combinations. For example, for the original OIE triple *("Michael Jordan"; "played for"; "Chicago Bulls")*, this process results in augmented triples such as *("Air Jordan"; "played for"; "Chicago Bulls")*, *("M.J."; "played for"; "The Bulls")*, etc.

## 4 OIE-to-KG Fact Linking Models

Our goal is to link surface-form ($"s"$; $"r"$; $"o"$) OIE triples, to canonical Knowledge Graph facts ($e_1$; $p$; $e_2$), where $e_1, e_2 \in \mathcal{E}$, and $p \in \mathcal{P}$. Each KG entity or predicate is represented as its surface-form KG label (e.g., "Michael Jordan"), and its KG-provided description (e.g., "American basketball player and businessman"). We henceforth refer to the entities and predicates as KG entries. Intuitively, since both data streams—the OIE triples and the KG facts—are in natural language, we opt to obtain their representations (i.e., embeddings) with a pre-trained language model. We decouple the OIE-to-KG linking in two steps: pre-ranking and re-ranking (see §4.1 and §4.2, respectively),[4] as is done commonly in entity retrieval (Wu et al., 2019) and image-text matching (Geigle et al., 2022; Li et al., 2021). See Fig. 2 for an overview and Appendix C for implementation details.

### 4.1 Pre-ranking OIE Slots to KG Entries

We denote this model as $\text{OIE}_{\text{ranker}}^{\text{pre}}$. It aims to generate OIE slot embeddings and KG entry embeddings, such that an OIE slot embedding yields higher similarity with its aligned KG entry's embedding compared to the other KG entries. Therefore, during training, we contrast the positive pairs against a set of negatives, thereby training the model to generate embeddings for a matching OIE slot and KG entry that lie close in the latent space. The motivation for such formulation is two-fold: (i) the number of entities is large, and could practically grow further, therefore computing the softmax over all KG entities during training is prohibitive; (ii) there may be unseen KG entries that we encounter during inference, therefore posing the problem as a standard classification inevitably leads to the model ignoring them. We use RoBERTa (Liu et al., 2019) to encode the OIE slots and the KG entries.

**OIE Embeddings.** We first add special tokens to indicate the start of each OIE slot: <SUBJ> for *"subject"*, <REL> for *"relation"*, <OBJ> for *"object"*. Hence, $t_1$ = *("M. Jordan"; "grew up in"; "Wilmington")* is represented as "<SUBJ> *M. Jordan* <REL> *grew up in* <OBJ> *Wilmington*". We then tokenize the OIE representation, denoted as $\hat{t}$,

---

[4]We perform the linking in roughly $O(|\mathcal{E}| + |\mathcal{P}| + |\mathcal{E}|)$ time complexity, where $|\mathcal{E}|$ and $|\mathcal{P}|$ are the number of KG entities and predicates respectively. Directly encoding whole KG facts would have a prohibitive complexity: In the limit, we could have $|\mathcal{E}| \times |\mathcal{P}| \times |\mathcal{E}|$ KG facts.

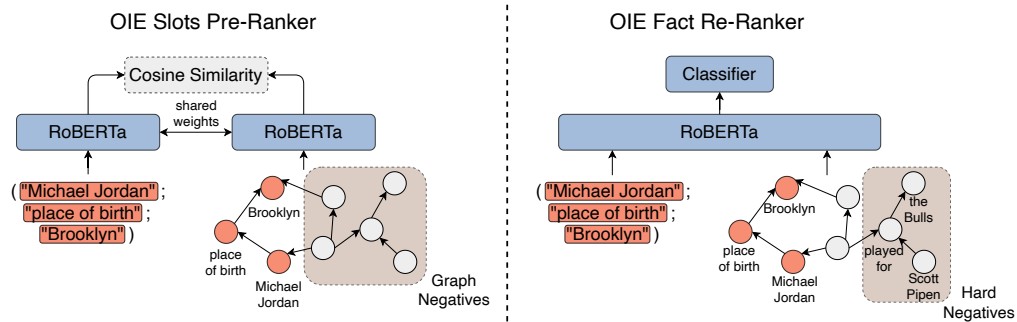

Figure 2: **Left:** $\text{OIE}^{\text{pre}}_{\text{ranker}}$ which performs pre-ranking of the OIE slots to KG entries independently (i.e., no context between them is considered). Trained with negatives sampled from the whole KG; **Right:** $\text{Fact}^{\text{re}}_{\text{ranker}}$ which attends between the whole OIE triple and KG fact to output their similarity; Trained with hard-negatives.

and obtain RoBERTa token embeddings. Finally, we pool *only* the special tokens' embeddings subsequently linearly projected in the desired latent space: $\hat{o}_i = \text{Linear}(\text{RoBERTa}(\hat{t}))$, where $\hat{o}$ is the slot embedding, and $i \in \mathbb{R}^3$ is the OIE slot index.

**KG Embeddings.** We represent both the entities and predicates as their label followed by their description (if available in the KG); e.g., the entity $(e)$ *Michael Jordan* is represented as "Michael Jordan <DESC> American basketball player and businessman...", and the predicate $(p)$ *place of birth* is represented as "place of birth <DESC> most specific known birth location of a person...". The <DESC> special token indicates the start of the description. We then tokenize the representation ($\hat{e}$ for entity, $\hat{p}$ for predicate), and obtain embeddings using the same RoBERTa model. We pool the <CLS> token embedding and linearly project it in the OIE representation space as: $\hat{k}_j = \text{Linear}(\text{RoBERTa}(b_j))$, where $j \in \{\hat{e}, \hat{p}\}$, $\hat{k}$ is the KG entry embedding, and $b$ is the tokenized KG entry representation.

**Linking OIEs to KG Facts.** Given an OIE slot embedding $o_i$ and a KG entry embedding $k_j$, we obtain their dot-product as: $\hat{s}_{pre} = o_i^T k_j$, where $o_i$ and $k_j$ are norm-scaled (i.e., $\hat{s}_{pre}$ represents the $\text{OIE}^{\text{pre}}_{\text{ranker}}$ cosine similarity). During inference, we link an OIE to a KG fact by selecting the most similar KG entry for each of the OIE slots.

**$\text{OIE}^{\text{pre}}_{\text{ranker}}$ Training.** We train the model using standard contrastive loss: we sample $N-1$ in-batch negative KG entries for each positive OIE slot $\leftrightarrow$ KG entry pair, where $N$ is the batch size. As per standard practice (van den Oord et al., 2018), we train the model using temperature-scaled InfoNCE contrastive loss: $\mathcal{L} = -\log \frac{e^{\hat{o}^T \hat{k}/\tau}}{e^{\hat{o}^T \hat{k}/\tau} + \sum_{n=1}^{N-1} e^{\hat{o}^T \hat{k}_n^-/\tau}}$,

where $\tau$ is the temperature. Note that during training, we only sample negative KG entry embeddings for each OIE slot, but not the other way around.

Sampling only in-batch negatives presents an issue as the training data represents only a limited subset of the whole KG (i.e., only the KG entries with paired OIE). During inference, however, we contrast each OIE slot against the whole KG to find the KG entry with which it exhibits the highest similarity. Therefore, for each OIE slot, we additionally sample $e$ negative entities and $p$ negative predicates at random from the whole KG (e.g., we would sample $\sum_{n=1}^{N-1+p} \hat{k}_n^-$ negative predicates).

## 4.2 Re-ranking OIEs to KG Facts

In certain scenarios (e.g., in the case of polysemous OIEs), matching whole OIEs with whole KG facts (i.e., not decoupled per OIE slot) could resolve the ambiguity and thus improve performance. To that end, for each OIE slot, we re-rank the $\text{OIE}^{\text{pre}}_{\text{ranker}}$ top-k most probable KG links. We denote this model as $\text{Fact}^{\text{re}}_{\text{ranker}}$. We perform self-attention between the OIE and the KG fact (both provided as input, separated by a <FACT> special token) with a single RoBERTa transformer, and return their similarity as: $\hat{s}_{re} = \sigma(\text{Linear}(\text{RoBERTa}(\hat{c})))$, where $\hat{c}$ is the concatenated OIE and KG fact representation, and $\hat{s}_{re}$ is the sigmoid ($\sigma$) normalized similarity.

**$\text{Fact}^{\text{re}}_{\text{ranker}}$ Training.** We train by sampling matching OIE $\leftrightarrow$ KG fact pairs as positives, and negatives, where we replace some KG fact slots (subject, predicate, object) with incorrect ones, generated as follows: We first obtain embeddings for each KG entry using the $\text{OIE}^{\text{pre}}_{\text{ranker}}$, and find its top-k most similar candidates w.r.t. all other KG entries. We then corrupt the ground truth KG fact by randomly sampling *only* from the top-k (hard)

negative candidates. Lastly, with 50% probability, we randomly mask (i.e., replace with a `<mask>` token) the description of the KG fact entries.

## 5 Experiments and Discussions

We measure accuracy to evaluate both OIE slot linking to KGs (§5.1), and Out-of-KG detection of OIE slots (§5.2). To measure OIE fact linking, we score a hit if *all* OIE slots are linked correctly. The error bars represent the standard error of the mean.

### 5.1 Linking OIEs to Knowledge Graphs

**Setup.** We explore the extent to which we can link OIE slots to a large-scale KG (Wikidata). We address two main research questions concerning the OIE-to-KG fact linking task: (i) To what extent do methods generalize to different KG entity facets? We consider transductive, inductive, or polysemous entities (see §2 for detailed definition); (ii) What is the performance impact of the KG size? We test two reference KG sizes: Benchmark Restricted KG ($\sim$650k entities, $\sim$0.6k predicates) and Large KG ($\sim$5.9M entities, $\sim$4k predicates).

**Methods.** We use the following methods to obtain results for the OIE slot linking task: (i) RANDOM: for each OIE slot, we sample a random KG entry; (ii) FREQUENCY: based on the training data statistics, we link each OIE slot to the most frequent KG entry (entity or predicate) in the training set; (iii) SIMCSE: we use a pretrained SimCSE model (Gao et al., 2021), where we represent each OIE slot and each KG entry (optionally, its description as well – if available in the KG) in a natural language format, and obtain their embeddings. We finally obtain the cosine similarity between each OIE slot and KG entry to perform the linking; (iv) $\text{OIE}_{\text{ranker}}^{\text{pre}}$ : We train a pre-ranking model as per the setup described in §4.1; (v) + Context: We append the context sentence to the OIE[5]; (vi) + $\text{Fact}_{\text{ranker}}^{\text{re}}$ : We additionally re-rank the top-k ($k = 3$) pre-ranked OIE slot links as per the setup in §4.2.

**Results.** In Table 1 we report the OIE-to-KG linking performance for each OIE slot, as well as linking on fact level. Overall, across the data splits, we observe that all unsupervised baselines perform poorly compared to models proposed in this work; indicating that OIE linking is not trivial, hence

off-the-shelf zero-shot models fail.[6] Additionally, there is a significant performance drop on the inductive and polysemous split compared to the transductive split, suggesting that the models are neither robust w.r.t. entities unseen during training, nor can cope with polysemous entities. Expectedly, leveraging extra context (via the sentence from where the OIE is obtained) aids the linking process in the inductive and polysemous split, as it helps generalization and disambiguation. Similarly, the $\text{Fact}_{\text{ranker}}^{\text{re}}$ brings a significant performance gain especially prominent for linking complete facts. Finally, we observe a significant performance impact of the KG size: across all splits, OIE-to-KG linking is more challenging on large-scale KG compared to the smaller Benchmark-Restricted KG.

**Training on Synthetic Data Improves Performance.** We explore the extent to which we can learn fact linking models using synthetic data. SynthIE (Josifoski et al., 2023) is a dataset that features natural language sentences paired with KG facts, where the sentences are obtained using a LLM. Namely, given a set of KG facts, Josifoski et al. (2023) prompt the LLM to generate a sentence which mentions (i.e., entails) all of the KG facts. Notably, if we can link OIE slots to KG facts by training on such synthetic dataset, then the KG remains the only human-annotated component for learning the OIE-to-KG fact linking task.

To measure to what extent we can link OIEs to KG facts using only synthetic data, we train $\text{OIE}_{\text{ranker}}^{\text{pre}}$ models on both REBEL and SynthIE, and report results (in Table 2) on inductive testing splits w.r.t. each dataset.[7] We observe that models trained on SynthIE are overall better OIE-to-KG fact linkers than models trained on REBEL (i.e., higher macro accuracy across datasets). This indicates that learning to link OIEs to KGs is possible using only synthetic data, thus the only human-annotated requirement remains to be a reference KG.

**Ablation Study: Importance of Entity Alias Augmentation.** We observe that in current datasets most surface form entities (in the natural language sentences) appear only with their "canonical" la-

---

[5]We structure the input as: "OIE `<SENT>` Sentence", where `<SENT>` is a special separator token.

[6]Note that simply linking the OIE relation to the most frequent KG predicate yields high accuracy, due to the KG predicates imbalance in REBEL.

[7]To preserve the inductive property, SynthIE's inductive test split contains only entities found in SynthIE that are not present in the REBEL train data.

| Method | Split Type | Subject | Relation | Object | Fact | Subject | Relation | Object | Fact |
|---|---|---|---|---|---|---|---|---|---|
| | | _Benchmark-Restricted Knowledge Graph_ | | | | _Large Knowledge Graph_ | | | |
| Random | Transductive | 0.0 ± 0.0 | 0.0 ± 0.0 | 0.0 ± 0.0 | 0.0 ± 0.0 | 0.0 ± 0.0 | 0.0 ± 0.0 | 0.0 ± 0.0 | 0.0 ± 0.0 |
| Frequency | Transductive | 0.0 ± 0.0 | 53.3 ± 0.1 | 8.1 ± 0.1 | 0.0 ± 0.0 | 0.0 ± 0.0 | 53.3 ± 0.1 | 8.1 ± 0.1 | 0.0 ± 0.0 |
| SimCSE | Transductive | 5.4 ± 0.1 | 0.0 ± 0.0 | 0.9 ± 0.1 | 0.0 ± 0.0 | 2.6 ± 0.1 | 0.0 ± 0.0 | 0.5 ± 0.0 | 0.0 ± 0.0 |
| $\text{OIE}_{\text{ranker}}^{\text{pre}}$ | Transductive | **86.8 ± 0.1** | **93.5 ± 0.1** | **95.7 ± 0.0** | **79.1 ± 0.1** | **78.7 ± 0.2** | **93.5 ± 0.1** | **93.1 ± 0.1** | **70.7 ± 0.2** |
| + Context | Transductive | 84.9 ± 0.1 | 92.2 ± 0.1 | 94.8 ± 0.1 | 77.7 ± 0.1 | 77.8 ± 0.1 | 92.2 ± 0.1 | 92.2 ± 0.1 | 70.2 ± 0.1 |
| Frequency | Inductive | 0.0 ± 0.0 | 0.1 ± 0.0 | 0.0 ± 0.0 | 0.0 ± 0.0 | 0.0 ± 0.0 | 0.1 ± 0.0 | 0.0 ± 0.0 | 0.0 ± 0.0 |
| SimCSE | Inductive | 12.6 ± 0.3 | 0.4 ± 0.1 | 6.8 ± 0.3 | 0.0 ± 0.0 | 8.1 ± 0.3 | 0.0 ± 0.0 | 3.2 ± 0.2 | 0.0 ± 0.0 |
| $\text{OIE}_{\text{ranker}}^{\text{pre}}$ | Inductive | 71.9 ± 0.4 | 69.8 ± 0.5 | 59.5 ± 0.5 | 34.9 ± 0.5 | 62.0 ± 0.5 | 69.8 ± 0.5 | 48.2 ± 0.5 | 25.4 ± 0.4 |
| + Context | Inductive | 74.5 ± 0.4 | **73.8 ± 0.4** | **62.3 ± 0.5** | 38.9 ± 0.5 | 64.5 ± 0.5 | **73.8 ± 0.4** | 50.8 ± 0.5 | 29.2 ± 0.5 |
| + $\text{Fact}_{\text{ranker}}^{\text{re}}$ | Inductive | **76.2 ± 0.4** | 67.6 ± 0.5 | 60.8 ± 0.5 | **40.6 ± 0.5** | **64.8 ± 0.5** | 66.5 ± 0.5 | **54.8 ± 0.5** | **32.9 ± 0.5** |
| Frequency | Polysemous | 0.0 ± 0.0 | 68.1 ± 0.4 | 11.7 ± 0.3 | 0.0 ± 0.0 | 0.0 ± 0.0 | 68.1 ± 0.4 | 11.7 ± 0.3 | 0.0 ± 0.0 |
| SimCSE | Polysemous | 1.3 ± 0.1 | 0.0 ± 0.0 | 0.3 ± 0.1 | 0.0 ± 0.0 | 0.4 ± 0.1 | 0.0 ± 0.0 | 0.0 ± 0.0 | 0.0 ± 0.0 |
| $\text{OIE}_{\text{ranker}}^{\text{pre}}$ | Polysemous | 68.8 ± 0.4 | 93.0 ± 0.2 | 94.6 ± 0.2 | 62.5 ± 0.4 | 58.2 ± 0.4 | 93.0 ± 0.2 | 92.4 ± 0.2 | 51.7 ± 0.4 |
| + Context | Polysemous | **77.9 ± 0.3** | **93.3 ± 0.2** | **95.8 ± 0.2** | **71.6 ± 0.4** | 69.5 ± 0.4 | **93.3 ± 0.2** | 94.0 ± 0.2 | 63.5 ± 0.4 |
| + $\text{Fact}_{\text{ranker}}^{\text{re}}$ | Polysemous | 75.3 ± 0.4 | 90.6 ± 0.2 | 95.3 ± 0.2 | 69.5 ± 0.4 | **74.1 ± 0.4** | 90.6 ± 0.2 | **95.6 ± 0.2** | **69.3 ± 0.4** |

Table 1: OIE slot and fact linking accuracy with models trained and evaluated on REBEL. Evaluation on a smaller Benchmark-Restricted KG, and a Large KG. The error bars indicate standard error of the mean.

| Train Data | Test Data | Subject | Relation | Object | Fact |
|---|---|---|---|---|---|
| REBEL | REBEL | 62.0 ± 0.5 | 69.8 ± 0.5 | 48.2 ± 0.5 | 25.4 ± 0.4 |
| REBEL | SynthIE | 53.6 ± 0.4 | 57.2 ± 0.4 | 44.9 ± 0.4 | 17.4 ± 0.3 |
| _Macro Score_ | | 57.8 | 63.5 | 46.6 | 21.4 |
| SynthIE | REBEL | 69.7 ± 0.3 | 63.2 ± 0.3 | 41.6 ± 0.3 | 22.4 ± 0.2 |
| SynthIE | SynthIE | 64.1 ± 0.6 | 81.2 ± 0.5 | 57.6 ± 0.6 | 34.5 ± 0.6 |
| _Macro Score_ | | **66.9** | **72.2** | **49.6** | **28.5** |

Table 2: Experiments on human-created (REBEL) and synthetically generated (SynthIE) datasets. OIE linking to a Large KG variant on inductive splits w.r.t. each dataset (see Appendix D.1 for details).

bel.[8] Since the OIEs represent surface-form facts, such lack of diversity prevents the models from learning more complex linking patterns. To overcome this, we perform entity alias augmentation in **FaLB** by adding the surface form aliases of the entities—available in Wikidata and manually curated—and ablate its impact on the OIE linking task. Besides $\text{OIE}_{\text{ranker}}^{\text{pre}}$ models trained on REBEL and SynthIE *with* entity alias augmentation, we train additional models *without* the alias augmented samples. We report results in Table 3 on inductive REBEL and SynthIE testing data, which *does* and *does not* feature entity aliases.

Expectedly, we observe that training with entity aliases allows us to link such OIE entity mentions more successfully than training without them. This was reflected on models trained on both REBEL

| Training Augmentation | Testing Augmentation | Subject | Relation | Object | Fact |
|---|---|---|---|---|---|
| | | _Models trained and evaluated on **REBEL**_ | | | |
| ✗ | ✗ | 90.7 ± 0.1 | 91.9 ± 0.1 | 87.1 ± 0.2 | 74.0 ± 0.2 |
| ✗ | ✓ | 61.6 ± 0.2 | 64.8 ± 0.2 | 41.5 ± 0.2 | 25.2 ± 0.2 |
| _Macro Score_ | | 76.2 | 78.4 | 64.3 | 49.6 |
| ✓ | ✗ | 89.0 ± 0.1 | 92.8 ± 0.1 | 85.1 ± 0.2 | 72.4 ± 0.2 |
| ✓ | ✓ | 79.0 ± 0.2 | 92.2 ± 0.1 | 89.0 ± 0.1 | 67.9 ± 0.2 |
| _Macro Score_ | | **84.0** | **92.5** | **87.1** | **70.15** |
| | | _Models trained and evaluated on **SynthIE**_ | | | |
| ✗ | ✗ | 91.8 ± 0.2 | 86.3 ± 0.2 | 90.7 ± 0.2 | 72.8 ± 0.3 |
| ✗ | ✓ | 61.2 ± 0.2 | 70.1 ± 0.2 | 40.7 ± 0.2 | 22.6 ± 0.2 |
| _Macro Score_ | | 76.5 | 78.2 | 65.7 | 47.7 |
| ✓ | ✗ | 91.1 ± 0.2 | 88.9 ± 0.2 | 90.5 ± 0.2 | 74.3 ± 0.3 |
| ✓ | ✓ | 80.1 ± 0.2 | 89.9 ± 0.1 | 86.3 ± 0.2 | 64.8 ± 0.2 |
| _Macro Score_ | | **85.6** | **89.4** | **88.4** | **69.6** |

Table 3: Impact of the **FaLB** entity alias augmentation step on REBEL and SynthIE. Inference is done on the full test set. The model is $\text{OIE}_{\text{ranker}}^{\text{pre}}$.

and SynthIE. We further observe that this step hurts the linking of specific OIE entity mentions only moderately, suggesting that OIE linking methods could be trained to be robust w.r.t. entity synsets. Finally, across all OIE slots and fact linking, the macro scores are significantly in favor of the model trained with entity aliases, on both datasets.

**Ablation Study: Importance of Fact-reranking.** We ablate the number of KG facts we rerank $(k)$ with $\text{Fact}_{\text{ranker}}^{\text{re}}$ and report results in Fig. 3. We observe the highest fact linking accuracy when performing reranking using the top-2 highest scoring KG entries for each OIE slot, although performance is within 1 standard deviation for $k = 2, 3, 4$. If reranking $k = 2$ facts, effectively, the $\text{Fact}_{\text{ranker}}^{\text{re}}$

---

[8]REBEL is constructed from Wikipedia abstracts, where the references use the canonical form name; SynthIE provides the KG fact (in text format) as is to the LLM, so naturally, the sentence generated does not feature the entity aliases. Therefore, *Michael Jordan*'s synonyms such as *M.J., Air Jordan* and *His Airness,* rarely appear in the data.

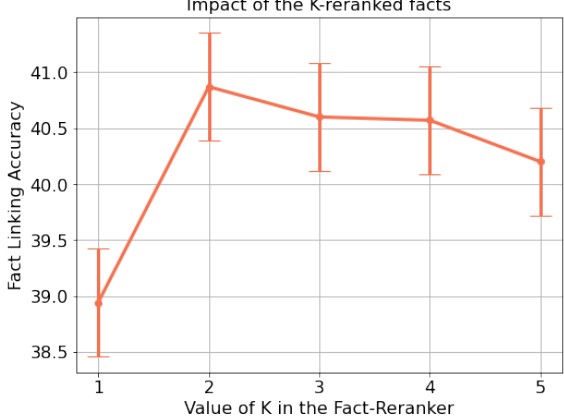

Figure 3: Ablating the impact of K in the $\text{Fact}^{\text{re}}_{\text{ranker}}$.

constructs $2^3$ OIE-KG fact pairs, reranks the list, and returns the highest scoring KG fact.

## 5.2 Detecting Out-of-Knowledge Graph OIEs

**Setup.** Prior works which study the OIE to KG linking problem (Zhang et al., 2019; Wood et al., 2021; Jiang et al., 2021) make the assumption that *all* OIE slots that need to be linked are present in the KG, which is rarely the case in practice. Here, we study (i) whether OIE slot linking methods can be converted to Out-of-KG detectors; (ii) the performance impact of the KG size on the Out-of-KG detection task; and (iii) to what extent is it more difficult to recognize the presence or absence of aliased OIE entity mentions.

**Datasets.** We create the Out-of-KG test split by imposing constraints such that we mimic an out-of-distribution setting. The constraints are: (i) both the KG entities and predicates referred by OIEs are unseen during training (i.e., not in the KG); (ii) the testing OIE-to-KG pairs do not come from the training data distribution: we use models trained on REBEL, but evaluate on SynthIE (which we use to create the Out-of-KG split). We measure Out-of-KG detection performance on the original OIEs, and OIEs with alias augmented entity slots.

**Evaluation Protocol.** We evaluate to what extent we can detect if an OIE slot refers to a concept outside of the KG. For each OIE slot, we add its corresponding concept (KG entity or predicate) to the KG, and score a hit if the model outputs a positive score for that slot. Conversely, for each OIE slot, we remove its corresponding entity or predicate from the KG, and score a hit if the model outputs a negative score. We report the averaged

accuracy over the two scenarios for each OIE slot.

**Methods.** We create three methods on top of $\text{OIE}^{\text{pre}}_{\text{ranker}}$ trained on REBEL: (i) CONFIDENCE@1-BASED HEURISTIC: We get the cosine similarities of the top-5 predictions from $\text{OIE}^{\text{pre}}_{\text{ranker}}$, compute the softmax, and threshold the top-1 confidence; (ii) ENTROPY-BASED HEURISTIC: We also obtain the top-5 cosine similarities, however, after softmax normalization compute their entropy. Finally, we threshold the entropy to obtain the prediction;[9] (iii) QUERY-KEY-VALUE CROSS-ATTENTION: We train a lightweight query-key-value cross-attention module on top of the frozen $\text{OIE}^{\text{pre}}_{\text{ranker}}$ embeddings. Given a query OIE slot embedding, the model attends over the KG entry embeddings representing the keys and values, and outputs a probability indicating the presence of the OIE slot in the KG (See Appendix E.1 for details).

**Results.** We report the Out-of-KG detection performance in Table 4. First, we observe that none of the models we evaluate are able to recognize whether an OIE relation has a corresponding KG predicate, indicating that models do not cope with zero-shot relations. Intuitively, the number of entities is orders of magnitude more than the relations, thus the models learn features which generalize to unseen data. On the other hand, the number of relations is limited ($\sim$600 during training) and therefore, the models overfit on this limited set. Second, similar to our observations on the OIE linking task, detection of Out-of-KG slots is significantly more difficult on data which features entity aliases, and even more difficult on larger KGs. Lastly, the best performing model is an entropy-based heuristic on top of the $\text{OIE}^{\text{pre}}_{\text{ranker}}$ output scores. Overall, our results indicate that Out-of-KG detection remains an open research problem.

## 6 Related Work

Prior work (Zhang et al., 2019; Wood et al., 2021; Jiang et al., 2021)—based on the ReVerb45k dataset (Vashishth et al., 2018)—considers fact linking as inductive and polysemous for the entities, but perform inductive inference for the relations. They also assume the OIE entities are linked to the KG entities a priori. In contrast, FaLB requires linking of all OIE slots to KG entries, with multiple evaluation facets (including the out-of-KG

---

[9]For each method, we find the optimal threshold on a hold-out set which we build on top of the REBEL validation set.

| Method | Subject | Relation | Object | Fact | Subject | Relation | Object | Fact |
|---|---|---|---|---|---|---|---|---|
| | Benchmark-Restricted Knowledge Graph | | | | Large Knowledge Graph | | | |
| *Testing data with entity alias augmentation* | | | | | | | | |
| Random | $50.0 \pm 1.3$ | $\mathbf{50.0 \pm 1.3}$ | $50.0 \pm 1.3$ | $12.5 \pm 0.8$ | $50.0 \pm 1.3$ | $\mathbf{50.0 \pm 1.3}$ | $50.0 \pm 1.3$ | $12.5 \pm 0.8$ |
| Confidence@1-based Heuristic | $63.8 \pm 1.2$ | $\mathbf{49.7 \pm 1.2}$ | $65.0 \pm 1.2$ | $21.5 \pm 0.4$ | $62.0 \pm 1.2$ | $\mathbf{49.7 \pm 1.2}$ | $62.7 \pm 1.2$ | $18.5 \pm 0.4$ |
| Entropy-based Heuristic | $\mathbf{67.7 \pm 1.2}$ | $49.0 \pm 1.2$ | $\mathbf{68.6 \pm 1.1}$ | $\mathbf{22.8 \pm 0.4}$ | $\mathbf{63.1 \pm 1.1}$ | $49.0 \pm 1.2$ | $\mathbf{64.5 \pm 1.1}$ | $\mathbf{23.0 \pm 0.4}$ |
| Query-Key-Value Cross-Attention | $62.4 \pm 1.2$ | $\mathbf{48.8 \pm 1.0}$ | $63.8 \pm 1.2$ | $17.2 \pm 0.2$ | $55.3 \pm 1.2$ | $\mathbf{49.2 \pm 1.2}$ | $56.8 \pm 1.2$ | $16.6 \pm 0.2$ |
| *Testing data without entity alias augmentation* | | | | | | | | |
| Random | $50.0 \pm 2.8$ | $\mathbf{50.0 \pm 2.8}$ | $50.0 \pm 2.8$ | $12.5 \pm 1.8$ | $50.0 \pm 1.3$ | $\mathbf{50.0 \pm 1.3}$ | $50.0 \pm 1.3$ | $12.5 \pm 1.8$ |
| Confidence@1-based Heuristic | $71.2 \pm 2.3$ | $\mathbf{49.6 \pm 2.6}$ | $72.1 \pm 2.3$ | $\mathbf{30.2 \pm 1.5}$ | $71.2 \pm 2.5$ | $\mathbf{49.6 \pm 2.7}$ | $69.1 \pm 2.5$ | $24.5 \pm 1.5$ |
| Entropy-based Heuristic | $\mathbf{77.5 \pm 2.3}$ | $49.0 \pm 2.7$ | $\mathbf{78.4 \pm 2.2}$ | $29.9 \pm 1.7$ | $\mathbf{73.3 \pm 2.3}$ | $49.1 \pm 2.7$ | $\mathbf{72.8 \pm 2.4}$ | $\mathbf{27.0 \pm 1.7}$ |
| Query-Key-Value Cross-Attention | $70.7 \pm 2.5$ | $\mathbf{49.6 \pm 2.7}$ | $70.9 \pm 2.5$ | $25.9 \pm 1.0$ | $59.8 \pm 2.7$ | $\mathbf{49.4 \pm 2.7}$ | $60.8 \pm 2.5$ | $19.7 \pm 0.9$ |

Table 4: Detection accuracy of Out-of-Knowledge Graph entities and predicates on the Out-of-KG split – built on top of SynthIE. All models trained on REBEL with entity alias augmentation.

| Dataset | Golden entities | Manually validated rels. | Multifaceted | Out-of-KG | Inductive | Transductive | Polysemous | # OIE Entities | # OIE Relations |
|---|---|---|---|---|---|---|---|---|---|
| FaLB (REBEL) | ✓ | ✓ | ✓ | ✓ | ✓ | ✓ | ✓ | 936,655 | 159,597 |
| FaLB (SynthIE) | ✓ | ✓ | ✓ | ✓ | ✓ | ✓ | ✓ | 1,049,922 | 147,056 |
| ReVerb45k | ✗ | ✗ | ✗ | ✗ | — | — | ✓ | 28,798 | 21,925 |

Table 5: Comparison of OIE-to-KG datasets: FaLB v.s. ReVerb45k. As transductivity and inductivity are defined only w.r.t. what the model has observed during training, we leave these entries blank, because ReVerb45k has only validation and testing dataset.

setup). In ReVerb45k (Vashishth et al., 2018) the links from the OIE entities to the KG entities are not *golden* (i.e., human labelled), but rather automatically obtained with an outdated entity linker (Lin et al., 2012); thus, the poor performance of the entity linker caps the performance of the fact linking models. In addition, the number of benchmark KG predicates is unrealistically small (only 250) compared to modern KGs (e.g., Wikidata). In this work, we mitigate these issues and build a benchmark (i) with golden links to KG entities that also (ii) reflects the size of modern KGs.

Furthermore, all prior work to date has relied on the strict assumption that *all* OIE surface form slots have a corresponding reference KG entity or predicate (Wu et al., 2019; Jiang et al., 2021; Zhang et al., 2019; Josifoski et al., 2022). This is obviously a false assumption, as any text corpus "in the wild" contains entities and predicates not present in even the largest of KGs (Gashteovski et al., 2020a). See App. A for detailed related work discussion.

Finally, most existing publicly available datasets do not address the problem of OIE-to-KG linking. Popular datasets like T-REx (Elsahar et al., 2018) and REBEL (Cabot and Navigli, 2021)–if considered without modification–address only Text-to-KG alignment, thus lack the OIE component. Therefore, these datasets are not directly comparable to FaLB. To the best of our knowledge, the only publicly-available OIE-to-KG dataset is Re-

Verb45k (Vashishth et al., 2018), which, as indicated above, has several drawbacks. For detailed comparisons with and FaLB, see Table 5.

# 7 Conclusion

We shed light on the OIE to KG linking problem, allowing us *to fuse the surface-form and opened-ended knowledge found in OIEs, with the canonical real-world KG facts*. We introduced a novel multifaceted benchmark which fixes prior work deficiencies, and proposed a set of task-specific baselines. Our experiments uncover that (i) linking inductive or polysemous OIEs to large KGs is challenging; (ii) we can learn OIE linking methods using only synthetic data; and (iii) detecting whether OIEs are Out-of-KG is an open research problem.

## Acknowledgements

We thank Dina Trajkovska for the help with the figures, and Mike Zhang for the fruitful discussions and feedback at the initial stages of the project.

## Limitations

Notably, the set of models we explore ignore the KG structure to obtain KG entry embeddings. Leveraging the underlying graph structure should, in theory, yield representations which generalize better to zero-shot samples (e.g., as is the case with detecting out-of-KG relations). Such KG entry embeddings could be even trained offline (i.e., as a separate step) with standard KG embedding methods (Bordes et al., 2013).

Last but not least, all data, resources, and models used in this work are specific to the English language. Notice however, our approach can be readily extended to languages other than English, while Wikipedia and Wikidata have versions in other languages – which we leave for future work.

## Ethical Impact

We are not aware of any direct ethical impact generated by our work. However, in general, care should be taken when applying our technology to sensitive use cases in high risk domains, such as healthcare.

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

## A Detailed Discussion on Related Work

In §2, we go over the problem statement, and in §6 we discussed how the different facets of our benchmark relate to prior and closely related work. Here, we provide a broader discussion where we group the related work based on the problem they address, and provide a more detailed discussion w.r.t. the differences with our work.

**Open Information Extraction (OIE).** OIE methods extract structured surface-form factual information from natural language text data, in the form of *("subject"; "relation"; "object")-*triples (Banko et al., 2007). Such systems are typically either rule-based (Mausam et al., 2012; Corro and Gemulla, 2013; Angeli et al., 2015; Gashteovski et al., 2017; Lauscher et al., 2019; Zopf and Gashteovski, 2023) or neural-based (Stanovsky et al., 2018; Hohenecker et al., 2020; Kotnis et al., 2022c,b; Bayat et al., 2022). Recent research showed that neural OIE systems still fall behind rule-based systems (Gashteovski et al., 2022a; Friedrich et al., 2022). To use the best of both worlds, in this work we use two rule-based systems: MinIE (Gashteovski et al., 2017) and Stanford OIE (Angeli et al., 2015); and two neural systems: MILIE (Kotnis et al., 2022c) and Multi$^2$OIE (Ro et al., 2020).

**Closed Information Extraction.** Given an input text, ClosedIE methods extract a set of (subject; relation; object)-triples where each triple can be expressed within the predefined schema—fixed sets of entities and predicates—of the reference KG (Josifoski et al., 2022; Trisedya et al., 2019; Sui et al., 2021; Josifoski et al., 2023). To ensure that each triple belongs to the KG, these methods prune candidates outside of the KG schema. Because of the pruning, these methods are unable to generate triples which contain entities or predicates that *are not already in the KG*. In turn, the task we address—OIE linking to KGs—allows us to both link OIE to existing KG candidates, as well as detect *novel* candidates which are outside of the KG schema.

**Open Information Extraction and Knowledge Graphs.** The context of OIE facts is used for many tasks for knowledge graphs, such as knowledge graph population (Broscheit et al., 2017; Lin et al., 2020; Zopf and Gashteovski, 2023), (open) link prediction (Broscheit et al., 2020; Kotnis et al., 2023), entity linking (Nanni et al., 2019) and entity

alignment (Friede and Gashteovski, 2022). Therefore, the task of OIE fact linking to KGs is of great importance, as OIE facts can provide knowledge that is outside of the KG schema, as well as knowledge that aligns with the KG schema (Gashteovski et al., 2020b). Several methods (Zhang et al., 2019; Jiang et al., 2021; Wood et al., 2021) address the problem of linking OIEs to KGs. These methods, however, make the assumption that the subject and object OIE slots (which link to KG entities) are linked *a priori*. These methods can link inductive OIE relations (outside of the training data), however, the assumption that the OIE entity slots are linked beforehand renders these methods not applicable for our task of linking (free-text) surface facts—also known as OIEs—to canonical large-scale KGs.

**Knowledge Graph Link Prediction.** These methods (Daza et al., 2021; Wang et al., 2021, 2022; Peng et al., 2022) address the problem of Knowledge Graph population by predicting the missing facts in the KG given only the current set of facts. They assign higher scores to valid KG facts (composed of entities and predicates currently in the graph), which need to be added in the graph, and lower scores to erroneous facts (i.e., wrong fact proposals). Even though, during inference, these methods deal with entities and predicates which were unseen during training, an implausible assumption is made that information is already extracted and canonicalized as a KG facts, which is never the case in practice.

**Multifaceted Evaluation.** NLP and KG tasks are typically evaluated on a held-out test set, by using evaluation frameworks that assign performance scores on a single value; e.g., accuracy (Petroni et al., 2020). In recent years, researchers have observed that such evaluation protocols are somewhat limited (Jain et al., 2023; Ye et al., 2021; Liu et al., 2021), because they do not expose the particular types of problems that the models might have. Hence, with such evaluation protocols, the tested models are more difficult to diagnose when they make errors (Ribeiro et al., 2020). Following prior work on multifaceted OIE (Gashteovski et al., 2022b) and multifaceted KG evaluation (Meilicke et al., 2018; Rim et al., 2021; Widjaja et al., 2022), we propose FaLB: a multifaceted evaluation framework for fact linking. FaLB allows fine-grained evaluation that helps users to pinpoint the source

of error (e.g., on which slot an examined model makes an error), which makes the benchmark more interpretable and useful for subsequent diagnostics of the models. In addition, our benchmark evaluates the performance of the models in different scenarios: inductive, transductive, polysemous, and out-of-KG. With this, the evaluation framework is more human-centric, in a sense that it can help users identify the model that they want for their needs instead of relying on single-score metrics (Kotnis et al., 2022a; Saralajew et al., 2022).

## B Benchmark Datasets Statistics

In Table 6 we report statistics of the benchmark datasets we create on top of REBEL and SynthIE, using the Wikidata Knowledge Graph.

## C Implementation Details

We train all models for 10 epochs using AdamW with a learning rate of 5e-5 and weight decay of 1e-3. We use a RoBERTa (Liu et al., 2019) model, distilled following the procedure of Sanh et al. (2019). The model consists of 6 layers, a hidden size of 768, and has 12 self-attention heads. To reduce the computational complexity and memory demands, we further linearly project the embeddings obtained from the RoBERTa model to a 200-dimensional latent space. When training the $OIE^{pre}_{ranker}$ models, we initialize the temperature $\tau$ to 0.07 as per Radford et al. (2021). We train $OIE^{pre}_{ranker}$ models with negatives that are sampled from the whole KG, where we sample 128 negative KG entities and 64 negative KG predicates. When training the $Fact^{re}_{ranker}$, we sample negatives such that we corrupt the slots of the KG fact by replacing them with incorrect ones. Instead of choosing the negatives at random, for each KG entry, we find its top-10 most similar KG entries, and sample negatives from this subset. This ensures that the $Fact^{re}_{ranker}$ model learns how to refine the predictions of the $OIE^{pre}_{ranker}$. We implement everything using PyTorch (Paszke et al., 2017), while we use HuggingFace transformers (Wolf et al., 2020) for the RoBERTa implementation. Lastly, we use Faiss (Johnson et al., 2019) to enable fast linking to Knowledge Graphs.

## D Discussion on the Quality of REBEL and SynthIE

While REBEL's entities are golden (i.e., obtained as the hyperlinks from Wikipedia abstracts which link to Wikipedia pages), the predicates between

them are obtained using a set of heuristics. This leads to imbalanced data, where most predicates occur only few times, and others occur orders of magnitude more. Consequently, such issue is reflected in the OIE-to-KG fact linking data that we obtain. To address this problem of REBEL, Josifoski et al. (2023) proposed SynthIE: a synthetically generated dataset which deals with the imbalance.

Josifoski et al. (2023) perform human evaluation to verify whether their synthetically generated dataset (of natural language sentences pair with KG facts) is of higher quality than REBEL. They obtain 44 data samples from REBEL, and generate a sentence using a Large Language Model (LLM) given only the KG triplets from the sample. Finally, they verify the triplet-set-to-text compatibility for both REBEL and SynthIE. Importantly, human evaluators find that the LLM generated sentences (i.e., the synthetic ones) are more compatible with the set of the KG facts (that is, the validity of the KG facts is higher in SynthIE compared to REBEL).

On the other hand, the issue we observe with SynthIE, is that due to the way the data is provided to LLM, the surface-form entities remain in their canonical text-form (i.e., their canonical denotation in the KG), and thus contain little variation. This diverges from the data that we find "in the wild". Critically, when people refer to entities in free-form natural language, they commonly use synonyms, aliases, abbreviations, nicknames, etc. For example, the former basketball player *"Michael Jordan"* could be referred to as *"Air Jordan"* and *"M.J."*.

To cope with this issue, we leverage the entity alias augmentation step in FaLB. By increasing the diversity of the OIE entity surface form, we are able to obtain a high quality OIE-to-KG fact linking synthetic dataset, thus the only human-annotated component remains to be the KG.

### D.1 Inductive Splits between Datasets

Importantly, the inductive evaluation is testing OIE linking to KG facts that consist of entities which were not part of the models' training data. In §5.1[10], we perform experiments by training models on REBEL and SynthIE, and then evaluate how well they perform on inductive data splits w.r.t. each of the datasets. Namely, to obtain an inductive REBEL testing split w.r.t. SynthIE, we find all testing samples from REBEL, which contain KG enti-

---

[10]More precisely, in the paragraph titled *"Results from Training on Synthetic Data"*

| Backbone Dataset | Split Type | # Total Samples | # Unique Entities | # Unique Predicates | # Unique Facts |
|---|---|---|---|---|---|
| REBEL | Training | 5,638,244 | 572,020 | 555 | 613,047 |
| SynthIE | Training | 7,749,603 | 685,959 | 757 | 766,032 |
| REBEL | Full Validation | 421,547 | 48,103 | 303 | 41,092 |
| SynthIE | Full Validation | 48,347 | 6,827 | 466 | 4,766 |
| REBEL | Full Testing split | 427,961 | 48,807 | 314 | 41,767 |
| SynthIE | Full Testing split | 240,300 | 28,660 | 635 | 22,661 |
| REBEL | Testing Transductivity | 241,995 | 14,730 | 192 | 13,496 |
| REBEL | Testing Inductivity | 10,300 | 4,423 | 185 | 2,297 |
| SynthIE | Testing Inductivity | 6,592 | 3,372 | 196 | 1,775 |
| REBEL | Testing Polysemy | 15,339 | 3,504 | 117 | 3,290 |
| SynthIE | Testing Out-of-KG | 1,604 | 443 | 106 | 264 |

Table 6: REBEL and SynthIE: overview of the number of samples in each dataset, number of unique KG entities, number of unique KG predicates, number of unique KG facts. Reported for all data splits of REBEL and SynthIE.

| Dataset Type | # Samples | # Entities | # Predicates | # Facts |
|---|---|---|---|---|
| REBEL w.r.t. Synthie | 26,937 | 8,781 | 181 | 5,304 |
| SynthIE w.r.t. REBEL | 17,931 | 6,318 | 488 | 3,499 |

Table 7: Overview of the total number of samples, number of unique KG entities, number of unique KG predicates, number of unique KG facts, and the. Reported for inductive splits w.r.t. each dataset.

ties that are *not part* of any SynthIE training samples. In turn, to obtain an inductive SynthIE testing split w.r.t. REBEL, we find all SynthIE testing samples which contain KG entities that are *not part* of any training samples from REBEL. We report statistics of the inductive splits w.r.t. each dataset in Table 7.

## E Details on the Out-of-Knowledge Graph Detection Task

In §5.2 we evaluate the ability of the models to detect whether an OIE slot (surface-form entity, or surface-form relation) is present in the Knowledge Graph. Intuitively, this task is more difficult than the OIE linking task, as the models need to generalize beyond the training data distribution to perform well on this task. Namely, when linking OIEs to a KG, all prior work (Zhang et al., 2019; Jiang et al., 2021; Wood et al., 2021) makes the conjecture that the testing data (in the open-world) is independent and identically distributed (i.i.d.) w.r.t. the training data, which is an invalid assumption in certain scenarios. Notably, the Out-of-KG OIEs lie outside of the training data distribution. Therefore, a model that performs well on the OIE linking task does not warrant high performance on the Out-of-KG detection task. We observed that this is the case

(i.e., a model performs well on the linking task, but performs poorly on the Out-of-KG task) when detecting out-of-KG relations.

Therefore, to ensure that the Out-of-KG data is indeed out-of-distribution (as it would be the case in practice), we impose the following constraints: (i) We select OIE-to-KG pairs from SynthIE, while the "backbone" model we build on top of is trained on REBEL; (ii) All entities and predicates—which are part of the KG facts from the testing data—are not in the KG at the time of training the $\text{OIE}^{\text{pre}}_{\text{ranker}}$.

To perform the evaluation, for each OIE slot we either leave its corresponding KG entry outside of the KG, and score a hit if the models predict a *negative score* for that slot; or, we perform imputation of the Out-of-KG entries (thus, they are now part of the KG), and score a hit if the models predict a *positive score* for that slot. Finally, we report the averaged accuracy over the two scenarios for each OIE slot. Note that, in this case both micro- and macro-average yield the same number, because we have the same number of samples for each scenario.

### E.1 Out-of-Knowledge Graph Detection Models

All models that we use for the Out-of-KG detection task in §5.2 are built on top of a $\text{OIE}^{\text{pre}}_{\text{ranker}}$, which is trained on REBEL. For all models, to obtain an out-of-KG indicator—True (1), or False (0)— we threshold the output of the models. For each model, we determine the optimal threshold (the confidence, or the entropy) on a hold-out validation dataset which we build on top of REBEL.

CONFIDENCE@1-BASED HEURISTIC: We obtain the KG links for each of the OIE slots using

the $\text{OIE}^{\text{pre}}_{\text{ranker}}$. The linking is characterized by the cosine similarity between the embeddings of each OIE slot and the KG entries. We then compute the softmax of the top-5 highest cosine similarities, and finally threshold the confidence@1 to obtain a prediction; such that a confidence $< T_c$ indicates an out-of-KG instance, and a confidence $> T_c$ indicates an instance inside the KG, where $T_c$ is the confidence threshold. We use $T_c = [0.235; 0.260; 0.235]$ for detecting out-of-KG subjects, relations and objects respectively.

**ENTROPY-BASED HEURISTIC:** Similar to the confidence@1-based method, we obtain the cosine similarities with the $\text{OIE}^{\text{pre}}_{\text{ranker}}$. However, instead of using the top-1 probability, we obtain the entropy of the top-5 predictions. We finally threshold the entropy to obtain a prediction, such that an entropy $> T_e$ indicates out-of-KG instance, and an entropy $< T_e$ indicates an instance inside the KG, where $T_e$ is the entropy threshold. We use $T_e = [1.60; 1.58; 1.60]$ for detecting out-of-KG subjects, relations and objects respectively.

**QUERY-KEY-VALUE CROSS-ATTENTION:** Using the $\text{OIE}^{\text{pre}}_{\text{ranker}}$ embeddings, we train a lightweight query-key-value cross-attention module on top with weights that are initialized with the identity matrix—at the start of training the $\text{OIE}^{\text{pre}}_{\text{ranker}}$ embeddings are used as is. Given a query OIE slot embedding, the model attends over the KG entry embeddings representing the keys and values, and outputs a sigmoid normalized score (i.e., a probability) indicating the presence confidence of the OIE slot in the KG. During training, to obtain in-batch negatives, for each OIE slot we drop the KG entry counterpart with 50% probability. We obtain additional negatives by sampling KG entries from the whole KG, which do not match any of the OIE slots in the batch. We train the model using the binary cross-entropy loss, such that, if the corresponding KG entry for an OIE slot is in the sampled KG subset, the model predicts a positive score averaged over the KG graph subset, and negative otherwise. We use a uniform threshold of $T_a = 0.3$ for all three slots.

## F  Data Quality

To assess FaLB's data quality, we performed manual human evaluation. In particular, we did the following steps:

1. We randomly selected 100 data points, where each data point contained information about the provenance sentence, a KG fact that is contained in the sentence, and a corresponding OIE surface fact that was extracted from the sentence.

2. Two expert annotators annotated each data point independently of whether the KG fact matches the OIE extraction semantically (see two labelled examples in Table 8).

3. We considered a data point as *"correct"* only if the two annotators agreed that the KG fact semantically matches the information in the OIE triple.

4. We computed accuracy, inter-annotator agreement and Kohen's kappa score.

We found that 97% of the data points are considered *"correct"* by both annotators. We also observed that the inter-annotator agreement was high: the annotators agreed in 99% of the cases, with high Kohen's kappa score (McHugh, 2012) of 0.80. Please refer to the supplementary material for the subset of samples which was provided to the expert annotators.

| KG fact (IDs) | KG fact (names) | OIE Triple |
|---|---|---|

**Sentence:** *"Hekimoğlu Ali Pasha Mosque was built between 1734–1735 in the Fatih district of Istanbul by Hekimoğlu Ali Pasha, who was born in Istanbul in 1689."*

(Q1584693; P19; Q406)  (Hekimoğlu Ali Pasha; place of birth; Istanbul)  *("H. Ali Pasha", "was born in", "Istanbul")*

**Label:** correct

**Sentence:** *"Pierre Hétu (April 22, 1936 in Montreal – December 3, 1998 in Montreal) was a conductor and pianist."*

(Q3385492, P19, Q340)  (Pierre Hétu; place of birth; Montreal)  *("Pierre Hétu", "April in", "Montreal")*

**Label:** incorrect

Table 8: Example annotations for FaLB data. The annotator sees: (1) the input sentence; (2) the KG fact with the original IDs (thus, the user can further check the meaning of the entity and predicate); (3) the surface text of the KG fact; (4) the OIE triple, extracted from the input sentence. The first matching is labelled as *"correct"*, because the KG fact semantically matches the OIE triple. The second matching is labelled as *"incorrect"*, due to the incorrect OIE relation *"April in"*.