# OpenReview forum: "Linking Surface Facts to Large-Scale Knowledge Graphs"
_EMNLP/2023/Conference — EMNLP 2023 Main_

### Official Review · Reviewer_WyCA · 2023-08-02

**Soundness:** 3

**Excitement:**

3: Ambivalent: It has merits (e.g., it reports state-of-the-art results, the idea is nice), but there are key weaknesses (e.g., it describes incremental work), and it can significantly benefit from another round of revision. However, I won't object to accepting it if my co-reviewers champion it.

**Paper Topic And Main Contributions:**

The authors introduce a new large-scale benchmark for OIE-to-KG linking. The benchmark assesses the performance of fact linking at a granular triple slot level. It also measures whether a system can recognize that a surfaceform does not have a match in the existing knowledge graph.

The benchmark defines multifaceted evaluation protocols and proposes several baselines. The evaluation focuses on:
- Transductive fact linking (linking entities and predicates seen during training)
- Inductive fact linking (test entities are not seen during training; measuring the generalization of fact linking models over entities)
 - Polysemous fact linking (entities are ambiguous)
 - Out-of-KG detection

The authors also show that, with the current baseline, it is possible to detect out-of-KG entities, but it is harder to detect out-of-KG relations.

**Questions For The Authors:**

Question A (line 078): Why is the setting where entity slots are a priori linked to KG entities unrealistic?
This could come from a pipeline system. The second system is just responsible for relation linking.

Question B: Is this setup of just linking triples without the context sentence realistic?

Question C (Section 3, High-precision OIE-KG Fact Alignments): The text suggests that if you match the subject and object of a triple with a subject and object of any KG fact then you consider it as a match. It means that you use all the predicates in KG between those entities. Aren't you using the already existing alignments mentioned in the Section 2 - Sentence-to-KG-dataset?

Question D: (Section 3, Data Augmentation to Increase Diversity): Do you also modify the original sentences with the given aliases? If not, then the future methods can easily learn shortcuts - if the triple subject/object doesn't occur in the source sentence then it's definitely in KG (because its alias was taken from KG).

Question E: (Section 3, Data Augmentation to Increase Diversity): What about pronouns or common nouns referring to the entity? The setup without them seems to be a bit unrealistic.

Question F (Section 4.1 Linking OIEs to KG Facts): In the embeddings you encode separately the information about subject, object and predicate. How do you deal with the contextual information then? If not, then for Michael Jordan slot entity "Michael Jordan basketball player" will be retrieved and there will be no room for disambiguation procedure.

Question G (Line 469): Does it mean you rerank 3^3 fact candidates (three entries per slot).

Question H: You report accuracy. Would precision/recall/f1 be more suitable to this task?

Question I: regarding SynthIE - If LLM is able to generate a sentence based on the fact, could it generate a fact based on a sentence? Should LLMs be considered as a baseline?

Question J (Section 5.1 Ablation study) "We observe that training with entity aliases allows us to link such OIE entity mentions more successfully than training without them" - where can it be observed? The scores on the non augmented test set get worse for subjects/objects.

**Reasons To Accept:**

- A novel benchmark for OIE-to-KG linking that covers multiple aspects of OIE to KG linking, as well as Out-of-KG detection.
 - Evaluation with multiple baselines.
 - Clear presentation.

**Reasons To Reject:**

- An artificial setup where only triple slots (subj, obj, pred) are used for linking without the use of the sentence in the context.
 - Weak baselines (that don't use the context sentence)
 - A data augmentation (paraphrasing / using different aliases for entities) that is applied on the triple level and not on the sentence level, which wouldn't allow to use source sentence (because the model could use the mismatch information between the surface form in the triple and the sentence)

**Reproducibility:**

4: Could mostly reproduce the results, but there may be some variation because of sample variance or minor variations in their interpretation of the protocol or method.

**Reviewer Confidence:**

3: Pretty sure, but there's a chance I missed something. Although I have a good feel for this area in general, I did not carefully check the paper's details, e.g., the math, experimental design, or novelty.

---

> ### Author Rebuttal · Authors · 2023-08-27
>
> We thank the review for an extensive and insightful review!
>
> We are, however, of the impression that the bulk of the reviewer's concerns (which we address with additional experiments and results reported below) relate to the baseline models which we evaluated and the fact that they do not leverage the sentential context of the OIE extraction. While the reviewer makes a very valid point -- which is why we additionally evaluated a baseline model that does leverage the sentence context -- we would like to emphasize that the baseline models are really not the main contribution of our work. Rather, it is the task (fact linking) and the comprehensive (and large) benchmark for evaluating linking models according to several facets that, crucially, reflect the usefulness of such models in real-world KG completion/augmentation applications (i.e., in particular the inductive and out-of-KG evaluation facets).
>
> Below we address the invididual points raised by the reviewer one by one.
>
> > An artificial setup where only triple slots (subj, relation, object) are used for linking without the use of the sentence in the context; Weak baselines (that don't use the context sentence).
>
> We would not describe this as an artificial setting. Indeed, the source sentence can be used to aid the linking, and, if such extra context is relevant to the task (may or may not be), it should lead to improved performance. Based on your suggestion, we train a model where we the source sentence (i.e., the sentence from where the OIE is obtained) is provided to the OIE pre-ranker model to generate OIE embeddings conditioned on the source sentence. Initially (in the experiments reported in the paper), the OIE is structured in the following way (L356 -- L367):
>
> <SUBJ> Michael Jordan <REL> grew up in <OBJ> Wilmington
>
> and is provided to RoBERTa to obtain OIE slot embeddings. Now, we add the source sentence together with another separator special token (<SENT>) in the following way:
>
> <SUBJ> Michael Jordan <REL> grew up in <OBJ> Wilmington <SENT> Michael Jordan is a famous basketball player who grew up in Wilmington.
>
> Then, as previously, we obtain OIE slot embeddings using the RoBERTa model. We report results of this sentence-augmented baseline below on our three (REBEL-based) OIE-to-KG fact linking splits: Transductive, Inductive, and Polysemous; and on the two versions of the the Knowledge Graph: Benchmark Restricted Knowledge Graph  (650k entities, 700 predicates) and Large Knowledge Graph (5M entities, 4k predicates).
>
> (OIE): Indicates just using the OIE to obtain the slot embeddings (as reported initially in the submitted paper)
> (OIE + Sent.): Indicates using the sentence context in addition to the OIE to obtain the slot embeddings
>
> |  | Benchmark Restricted Knowledge Graph | Large Knowledge Graph |
> |---|---|---|
> | Transductive (OIE) | 79.1 ± 0.1 | 70.7 ± 0.2 |
> | Transductive (OIE + Sent.) | 76.3 ± 0.2 | 68.2 ± 0.21 |
> | Inductive (OIE) | 34.9 ± 0.5 | 25.4 ± 0.4 |
> | Inductive (OIE + Sent.) | 37.4 ± 0.5 | 27.14 ± 0.44 |
> | Polysemous (OIE) | 62.5 ± 0.4 | 51.7 ± 0.4 |
> | Polysemous (OIE + Sent.) | 70.0 ± 0.37 | 62.5 ± 0.4 |
>
> From the reported results we can conclude that, leveraging the sentence context helps linking inductive and especially polysemous entity mentions, but not in the transductive task. In those cases, the context provided by the source sentence aids the linking performance, as it, expectedly, helps generalization and disambiguation. In the case of transductive OIEs, the sentence context hurts the performance as these OIE mentions have already been observed during training -- the sentential context here presumably "noisifies" the representations for these entities observed in training. We will add this baseline (OIE + Sent.) in Table 1.
>
> We would, however, like to emphasize once mode that the baseline models are not really the central contribution of our work: it is the multi-faceted evaluation benchmark for fact linking (the contribution that the reviewer is appreciative of, as far as we can tell from the review listed reasons for acceptance).
>
> > Data augmentation (paraphrasing / using different aliases for entities) applied on the triple level and not on the sentence level, which wouldn't allow to use source sentence (because the model could use the mismatch information between the surface form in the triple and the sentence)
>
> We do not think that this is correct. Namely, in the experimental results reported above, we perform the entity alias augmentation on both the triple level and the corresponding sentence added to the context (after the special <SENT> token). From an implementation standpoint, we locate both entities in the source sentence (via regex matching), and simply replace them with their aliases in both the OIE triple and the source sentence.
>
> > Question A: Why is the setting where entity slots are a priori linked to KG entities unrealistic? This could come from a pipeline system. The second system is just responsible for relation linking.
>
> This is realistic for a model, it is not realistic (or perhaps a more suitable word here is "appropriate") as a starting point for creating a fact-linking benchmark, if the entity linking has been carried out automatically by an (imperfect) entity linking model (and there exist no large-scale manually entity-linked datasets). Erroneously linked entities leads to erroneous facts -- evaluating fact linking models  on such facts is then misleading and not really indicative of models' true performance. (see also reply to to R1 regarding the ReVerb45K dataset).
>
> > Question B: Is this setup of just linking triples without the context sentence realistic?
>
> The main contribution of the paper is the comprehensive multi-faceted benchmark that allows one to evaluate both models that leverage only the information in the OIE triples (as we did in the original submission) as well as those that additionally leverage the sentence from which the OIE triple is extracted (as we did in the additional experiment reported above). As the results above show, having sentential context is not always beneficial, it depends on the task -- it does not help, for example, in transductive linking, but is beneficial in inductive and especially polysemous linking.
>
> Arguably, there are many different ways in which one could leverage/incorporate the sentence information in the fact-linking model: it was not our aim to extensively evaluate a wide range of plausible models but rather provide a most comprehensive benchmark to date (one that tests the linking models for things that existing benchmarks do not, but which are critical in real-world application of those models) so and offer it to the research community to catalyze efforts towards more robust fact linking models.
>
> > Question C (Section 3, High-precision OIE-KG Fact Alignments): The text suggests that if you match the subject and object of a triple with a subject and object of any KG fact then you consider it as a match. It means that you use all the predicates in KG between those entities. Aren't you using the already existing alignments mentioned in the Section 2 - Sentence-to-KG-dataset?
>
> We need to do high-precision OIE-to-KG, not sentence-to-KG (i.e., we need to know exactly to which KG entities the OIE slots correspond to; yet we start from a dataset in which the KG entities are assigned to sentences). This is why it is crucial to ensure that the OIE triples are of high quality.
>
> > Question D: (Section 3, Data Augmentation to Increase Diversity): Do you also modify the original sentences with the given aliases? If not, then the future methods can easily learn shortcuts - if the triple subject/object doesn't occur in the source sentence then it's definitely in KG (because its alias was taken from KG).
>
> Yes we do, as we stated above: we perform the entity alias augmentation on both the triple level and the (context) sentence level.
>
> > Question E: (Section 3, Data Augmentation to Increase Diversity): What about pronouns or common nouns referring to the entity? The setup without them seems to be a bit unrealistic.
>
> If the pronouns and common nouns are present in the data, then we include them. That being said, we do include common nouns, because they appear as Wikidata aliases (e.g., our links are also for concepts such as "train", not just named entities like names of people). Pronouns are not present here, because they are not part of the manually curated data (Wikidata). If we artificially add pronouns and common nouns, we introduce bias and reduce the quality of our data.
>
> > Question F (Section 4.1 Linking OIEs to KG Facts): In the embeddings you encode separately the information about subject, object, and predicate. How do you deal with the contextual information then? If not, then for Michael Jordan slot entity "Michael Jordan basketball player" will be retrieved and there will be no room for a disambiguation procedure.
>
> We obtain a separate embedding for each OIE slot, where each OIE slot embedding (subj, relation or object) is contextualized by the other. In other words, the entire triple is the input to the RoBERTa encoder, so tokens of each slot are contextualized with the other two slots (this is detailed in L356-367).
>
> > Question G (Line 469): Does it mean you rerank 3^3 fact candidates (three entries per slot).
>
> Yes. We retrieve the top-3 candidates for each OIE slot, and construct all possible OIE-KG fact combinations (3^3).
>
> > Question H: Would precision/recall/f1 be more suitable to this task instead of accuracy?
>
> We follow standard practice of entity linking methods [1] and report accuracy.
>
> > Question I: In SynthIE, If an LLM is able to generate a sentence based on the fact, could it generate a fact based on a sentence? Should LLMs be considered as a baseline?
>
> We do not see a straightforward way to use an LLM for OIE-to-KG fact linking (or sentence-to-KG fact linking), as the LLM would need to be able to access to the entire KG: this amount of data by far exceeds what even the largest LLMs could store in their context. In fact, the main premise of SynthIE [2] is that LLMs can generate synthetic data even for tasks that cannot be solved directly by the LLM. They show that, for problems with structured outputs, an LLM can be prompted to perform the task in the opposite direction (i.e., to generate plausible text for the target structure).
>
> > Question J (Section 5.1 Ablation study) "We observe that training with entity aliases allows us to link such OIE entity mentions more successfully than training without them" - where can it be observed? The scores on the non augmented test set get worse for subjects/objects.
>
> We apologize for our ambiguous wording. We here refer to the results from Table 3 where we only evaluate on entity aliases (numbers on REBEL, same trend can be observed on SynthIE):
>
> | Training aliases | Testing aliases | Subject | Relation | Object | Fact |
> |---|---|---|---|---|---|
> | ✗ | ✓ | 61.6 ± 0.2 | 64.8 ± 0.2 | 41.5 ± 0.2 | 25.2 ± 0.2 |
> | ✓ | ✓ | 79.0 ± 0.2 | 92.2 ± 0.1 | 89.0 ± 0.1 | 67.9 ± 0.2 |
>
> Indeed, training with entity aliases (row 2) allows us to link such (aliased) OIE entity mentions more successfully than if we perform training without entity aliases (row 1).
>
> ### References
>
> [1] We et al., 2019. Scalable Zero-shot Entity Linking with Dense Entity Retrieval
>
> [2] Josifovski et al., 2023. Exploiting asymmetry for synthetic training data generation: Synthie and the case of information extraction

---

### Official Review · Reviewer_PQB3 · 2023-08-03

**Soundness:** 4

**Excitement:**

3: Ambivalent: It has merits (e.g., it reports state-of-the-art results, the idea is nice), but there are key weaknesses (e.g., it describes incremental work), and it can significantly benefit from another round of revision. However, I won't object to accepting it if my co-reviewers champion it.

**Missing References:**

I didn’t find any missing references.

**Paper Topic And Main Contributions:**

To bridge the gap between the schema-free surface facts extracted from text and schema-fixed KG knowledge., the authors move away from unrealistic and incomplete assumptions of prior work and proposed a large-scale dataset for OIE-to-KG linking, multifaceted evaluation protocols that cover all aspects of linking OIE facts to KGs, and OIE-to-KG fact linking models. Extensive computational experiments on the dataset with the several evaluations showed the effectiveness of the framework.

Strengths:
1.Sufficient technical details, showing the transformation process;
2.Extensive computational experiments results showed the effectiveness of the framework;
3.The paper is very well written with clarity, easy-to-follow logic, and adequate discussions.


Weaknesses:
1.The novelty is somewhat limited, making the model is a combination of existing works;
2.The experimental results in Table 1 are somewhat unbelievable.

**Questions For The Authors:**

What is the main novelty of the framework(fig. 2)?

**Reasons To Accept:**

The paper gave a valuable dataset for OIE-to-KG linking. It would also be beneficial for those who considering the OIE-to-KG linking problem.

**Reasons To Reject:**

The novelty is somewhat limited, making the model is a combination of existing works.

**Reproducibility:**

4: Could mostly reproduce the results, but there may be some variation because of sample variance or minor variations in their interpretation of the protocol or method.

**Reviewer Confidence:**

3: Pretty sure, but there's a chance I missed something. Although I have a good feel for this area in general, I did not carefully check the paper's details, e.g., the math, experimental design, or novelty.

**Typos Grammar Style And Presentation Improvements:**

The structure of this paper is complete and English writing skills used in the paper is good. I didn’t find any obvious writing problems in the paper.

---

> ### Author Rebuttal · Authors · 2023-08-27
>
> >Limited novelty of the paper
>
> and
>
> >Main novelty of the framework (Fig. 2)?
>
> The main contributions (i.e., novelty) of our work is in the following: (1) proposed task -- fact linking, including four mutually complementary evaluation facets (transductive, inductive, polysemous, Out-of-KG) that allow for a comprehensive evaluation and profiling of OIE-to-KG linking models, (2) the proposed and created benchmark that allows to actually evaluate the usefulness of linking models for real-world applications targeting KGs augmentation. The implemented baseline model(s) do not represent the central contribution of this work, i.e., the goal of the model itself is not be be novel, but to represent a solid baseline on the new task(s)/benchmark: *we propose a novel benchmark dataset*, not a novel model. The strong (baseline) models we propose for this task are based on the pre-rank & re-rank paradigm, which are commonly used in image-text matching and retrieval.
>
> >Experimental results in Table 1 are somewhat unbelievable.
>
> Could the reviewer be more specific? What in particular is "unbelievable"? The results in Table 1 report the OIE-to-KG facts linking accuracy for different aspects of our benchmark and for a series of models. The experimental setup, a discussion of the results, and their implication are detailed in Section 5.1.

---

### Official Review · Reviewer_kopq · 2023-08-04

**Soundness:** 4

**Excitement:**

3: Ambivalent: It has merits (e.g., it reports state-of-the-art results, the idea is nice), but there are key weaknesses (e.g., it describes incremental work), and it can significantly benefit from another round of revision. However, I won't object to accepting it if my co-reviewers champion it.

**Missing References:**

1. KBPearl: A Knowledge Base Population System Supported by Joint Entity and Relation Linking, Lin et al., 2020
2. CESI: Canonicalizing Open Knowledge Bases using Embeddings and Side Information, Vashisth et al., 2019


**Paper Topic And Main Contributions:**

The paper curates a new data set for the task of extracting canonicalized facts from text. The proposed method uses existing Open IE techniques to extract triples from sentences and maps them to a target knowledge graph (Wikidata). The data curation process includes transductive (entities and relations are seen) and inductive settings (some of entities and relations are unseen) as well as  polysemous and out-of-KG entities. The paper also proposes a model that maps OIE triples to KG facts.

**Questions For The Authors:**

1. What is the usefulness of reporting the slot linking performance? If the end task is to extract KG facts from text, does it matter if the model can correctly link one of the slots while incorrectly linking on the other slots?

2. Why does SimCSE perform so poorly? It is mentioned in the paper that the OIE triples and KG facts were converted to sentences. How was this conversion done? Was  any LLM used for this?


**Reasons To Accept:**

1. The proposed data set can be used as a new benchmark for KG fact extraction from text.
2. The dataset curation process considers a wide variety of settings which can serve a test bed for existing and future models for this task.
3. The paper proposes a solution that works well on the proposed data set.


**Reasons To Reject:**

1. No clear comparison is made between the proposed data set and other existing large scale datasets (e.g., T-Rex, ReVerb45K, etc.). Such comparisons would make it clear why the community should adopt this data set instead of existing ones.
2. Lack of strong baselines. The proposed method is overwhelmingly better than any of the baselines. There are existing works of fact extraction from text (e.g., KBPearl, CESI) that could serve as a strong baseline.


**Reproducibility:**

3: Could reproduce the results with some difficulty. The settings of parameters are underspecified or subjectively determined; the training/evaluation data are not widely available.

**Reviewer Confidence:**

4: Quite sure. I tried to check the important points carefully. It's unlikely, though conceivable, that I missed something that should affect my ratings.

**Typos Grammar Style And Presentation Improvements:**

* L250-L261 can be presented as a table.

---

> ### Author Rebuttal · Authors · 2023-08-27
>
> We thank the review for the extensive review and insightful comments!
>
> > No clear comparison between the proposed dataset and existing large-scale datasets (e.g., T-Rex, ReVerb45K, etc.)
>
> We thank the reviewer for the suggestion. We do make a comparison between these datasets and ours in the paper, though perhaps this is not as clearly articulated, as the reviewer suggests. Explicitly, our benchmark is superior to ReVerb45K in the following aspects:
>
> 1. In ReVerb45k, the links of the OIE entities to KG entities are *not golden* (i.e., human validated), but rather automatically obtained with a very outdated entity linker [1]. Therefore, as we state in the paper (L178 -- L183): the ``poor performance of the entity linker caps the fact linking models' performance'' on ReVerb45k.
>
> 2. The number of unique OIE entities & relations in ReVerb45k is drastically smaller than our benchmark, making our dataset a much more challenging benchmark for fact linking models.
>
> |           | # OIE Entities | # OIE relations |
> |-----------|----------------|-----------------|
> | ReVerb45k | 28k            | 22k             |
> | Ours      | 936k           | 160k            |
> |           |                |                 |
>
>
> 3. ReVerb45k provides a single-facet of OIE evaluation: in which different types/sources of errors are conflated. In contrast, our benchmark decouples these and allows for evaluation of mutually very different OIE-to-KG linking scenarios: Transductive, Inductive, Polysemous, (and Out-of-KG, not supported by ReVerb45k).
>
> 4. ReVerb45k is based a confidence threshold of an automatic aligner, while our data is manually validated by two independent annotators -- inter-annotator agreement of 99%; Kohen’s kappa of 0.80 (L291);
>
> 5. In ReVerb45k there is no Out-of-KG testing data, which makes it inadequate to test a scenario of immense practical value: recognizing OIE extractions that correspond to facts consisting of entities and/or relations that are not in the KG at all.
>
> In a similaer fashion, T-Rex differs from our benchmark in the following:
>
> 1. T-Rex does not support neither the inductive evaluation, nor the Out-of-KG evaluation, both critical for the applicability of fact linking models and systems in real-world applications (where entities and/or relations for which the predictions need to be made were not seen in model training or are not present in the KG at all). In other words, the fact linking performance on T-Rex does not really measure models'  usefulness "in the wild".
>
> 2. The dataset is primarily constructed for text-to-KG generation, while our dataset is meant to link OIEs to large-scale KGs.
>
> We will articulate these critical differences w.r.t. to both datasets more clearly in the final version of the paper, perhaps in a tabular overview.
>
> > Lack of strong baselines: the method is overwhelmingly better than the baselines. There are existing works (e.g., KBPearl, CESI) that could serve as a strong baseline.
>
> We agree that KBPearl would be a nice baseline to compare against. Unfortunately, the KBPearl code is not publicly available and, we therefore could not compare it on our benchmark. As for CESI, it solves a different problem, which makes it incompatible to our study. In particular, CESI deals with entity and relation *canonicalization*, and not *linking*. That means that if there are entity mentions like "Joe Biden", "Joseph Biden" and "46th President of USA", CESI should be able to group them into an entity synset. CESI, however, is not able to link these entity synsets to a unique KG entity. In the final version of the paper, we will include these references and explain these differences clearly.
>
> > Usefulness of reporting the OIE slot linking performance? The end task is extracting KG facts from text, does it matter if the model can correctly link one of the slots while incorrectly linking on the other slots?
>
> The goal is to pinpoint the source of error, which makes the benchmark more interpretable and useful for subsequent diagnostics. For example, Gashteovski et al. 2022 [2] showed that OIE systems struggle with extracting objects. Many other NLP benchmarks aim at such fine-grained measurements as well [3, 4, 5, 6], because they provide insights on models' failures. Relying on single-score metrics (e.g., only fact-level accuracy) limits the analysis of the sources of models' errors.
>
> > Poor performance of SimCSE?
>
> SimCSE is an unsupervised sentence embedding model. On the other hand, the OIE-to-KG fact linking task requires methods to link an OIE mention (entity or relationship) with a large-scale KG that contains 5M entities and 4k predicates (e.g., in Wikidata). Because of that, models not explicitly trained for this task (including SimCSE) are suboptimal.
>
> ### References
>
> [1] Lin et al., 2012. Entity Linking at Web Scale. https://aclanthology.org/W12-3016/
>
> [2] Gashteovski et al., 2022. BenchIE: A Framework for Multi-Faceted Fact-Based Open Information Extraction Evaluation. https://aclanthology.org/2022.acl-long.307/
>
> [3] Hupkes et al., 2022. State-of-the-art generalisation research in NLP: A taxonomy and review https://arxiv.org/pdf/2210.03050.pdf
>
> [4] Liu et al., 2021. ExplainaBoard: An Explainable Leaderboard for NLP. https://aclanthology.org/2021.acl-demo.34/
>
> [5] Ye et al., 2021. Towards More Fine-grained and Reliable NLP Performance Prediction. https://aclanthology.org/2021.eacl-main.324/
>
> [6] Fu et al., 2020. Interpretable Multi-dataset Evaluation for Named Entity Recognition. https://aclanthology.org/2020.emnlp-main.489/

---

### Meta-Review · Area_Chair_9YWT · 2023-09-14

**Recommendation:** 4

**Metareview:**

The paper presents a new benchmark (dataset and evaluations) for OIE fact linking. The main focus of the benchmark is the different aspects of OIE linking (seen vs unseen vs polysemous entities), as well as out-of-KG detection.

All three reviewers find the proposed benchmark as valuable resource and, as Reviewer kopq points out, the dataset curation process itself is an interesting contribution. The lack of extensive comparison to other existing datasets was brought up as an issue, but this has been addressed by the authors and should be included in the revised paper.

All three viewers pointed to the lack of strong baselines as an issue, with Reviewer WyCA explicitly calling out the lack of models that used the context sentence as input. While I agree with the authors that the benchmark itself and not the baselines are the main contribution of the paper, I think that using the strongest baselines possible would better demonstrate the usefulness of the resource. In response to Reviewer WyCA, the authors have included an additional baseline (that uses the context sentence) and this - in part - addresses the reviewers' main concern. The issues regarding the novelty (presumably of the baselines) raised by reviewer PQB3 were not backed up with further clarifications and were not considered further.

---

### Decision · Program_Chairs · 2023-10-07

**Decision:**

Accept-Main

**Comment:**

The paper presents a new benchmark (dataset and evaluations) for OIE fact linking. The main focus of the benchmark is the different aspects of OIE linking (seen vs unseen vs polysemous entities), as well as out-of-KG detection.

All three reviewers find the proposed benchmark as valuable resource and, as Reviewer kopq points out, the dataset curation process itself is an interesting contribution. The lack of extensive comparison to other existing datasets was brought up as an issue, but this has been addressed by the authors and should be included in the revised paper.

All three viewers pointed to the lack of strong baselines as an issue, with Reviewer WyCA explicitly calling out the lack of models that used the context sentence as input. While I agree with the authors that the benchmark itself and not the baselines are the main contribution of the paper, I think that using the strongest baselines possible would better demonstrate the usefulness of the resource. In response to Reviewer WyCA, the authors have included an additional baseline (that uses the context sentence) and this - in part - addresses the reviewers' main concern. The issues regarding the novelty (presumably of the baselines) raised by reviewer PQB3 were not backed up with further clarifications and were not considered further.